# Autophagy Triggers Tamoxifen Resistance in Human Breast Cancer Cells by Preventing Drug-Induced Lysosomal Damage

**DOI:** 10.3390/cancers13061252

**Published:** 2021-03-12

**Authors:** Chiara Actis, Giuliana Muzio, Riccardo Autelli

**Affiliations:** Department of Clinical and Biological Sciences, University of Turin, 10125 Turin, Italy; chiara.actis@unito.it (C.A.); giuliana.muzio@unito.it (G.M.)

**Keywords:** breast cancer, endocrine resistance, autophagy, lysophagy, lysosomal membrane permeabilization, iron-binding proteins

## Abstract

**Simple Summary:**

Endocrine therapy with tamoxifen or other endocrine drugs represents the standard treatment for estrogen receptor-positive breast cancer. In spite of effectiveness of this therapy, onset of drug resistance worsens the prognosis of about 30% of patients. Autophagy has recently been proposed as a key player of drug resistance, but the underlying mechanisms are not completely understood. In this research, the authors investigate how autophagy triggers drug resistance in breast cancer cells. The results evidence that tamoxifen affects lysosome integrity, which suggests that this effect may contribute to the anticancer activity of this drug. Activation of autophagy and overexpression of iron-binding proteins synergize in protecting the lysosomal compartment, restraining drug effectiveness in breast cancer cells. According to these results, tamoxifen-resistant cells show an increased autophagic flux and overexpress iron-binding proteins. These findings indicate that screening for the level of iron-binding proteins may help to identify patients at risk for developing drug resistance.

**Abstract:**

Endocrine resistance is a major complication during treatment of estrogen receptor-positive breast cancer. Although autophagy has recently gained increasing consideration among the causative factors, the link between autophagy and endocrine resistance remains elusive. Here, we investigate the autophagy-based mechanisms of tamoxifen resistance in MCF7 cells. Tamoxifen (Tam) triggers autophagy and affects the lysosomal compartment of MCF7 cells, such that activated autophagy supports disposal of tamoxifen-damaged lysosomes by lysophagy. MCF7 cells resistant to 5 µM tamoxifen (MCF7-TamR) have a higher autophagic flux and an enhanced resistance to Tam-induced lysosomal alterations compared to parental cells, which suggests a correlation between the two events. MCF7-TamR cells overexpress messenger RNAs (mRNAs) for metallothionein 2A and ferritin heavy chain, and they are re-sensitized to Tam by inhibition of autophagy. Overexpressing these proteins in parental MCF7 cells protects lysosomes from Tam-induced damage and preserves viability, while inhibiting autophagy abrogates lysosome protection. Consistently, we also demonstrate that other breast cancer cells that overexpress selected mRNAs encoding iron-binding proteins are less sensitive to Tam-induced lysosomal damage when autophagy is activated. Collectively, our data demonstrate that autophagy triggers Tam resistance in breast cancer cells by favoring the lysosomal relocation of overexpressed factors that restrain tamoxifen-induced lysosomal damage.

## 1. Introduction

Breast cancer is the most common malignancy among women in Western countries, and the second leading cause of cancer-related death in women worldwide. Breast cancer is a heterogeneous disease currently classified in five subtypes according to the expression of critical proteins, such as hormone receptors, growth factor receptors, proliferation markers, or none [1]. Approximately 70% of newly diagnosed breast cancer express the estrogen receptor (ER) α or β; its presence has primed the development of a number of drugs aimed at interfering with estrogen synthesis, their binding to the receptor, or intracellular ER signaling, in a therapeutic approach designated “endocrine therapy”. Among the several endocrine drugs, tamoxifen (Tam), aromatase inhibitors, and other selective ER modulators (SERMs) or downregulators (SERDs) represent the standard treatment for ER^+^ breast cancer. This approach has shown great clinical effectiveness, but intrinsic (also called de novo) or acquired drug resistance in approximately 30% of ER^+^ breast cancer patients complicates the prognosis and accounts for disease progression and reduced survival. The mechanisms underlying drug resistance of breast cancer have been extensively investigated and partly elucidated [2,3]. Lack of expression or mutations of ER was initially implicated as the most likely cause of endocrine resistance. However, recently dysregulation of autophagy has emerged as a prominent mechanism contributing to the resistance of breast cancer cells to anticancer drugs [4]. Autophagy is a phylogenetically conserved catabolic process triggered by nutrient restriction or cellular stresses (for a review see [5]). Autophagy partakes in the homeostatic control of normal cell populations, as well as in cancer prevention, onset, and progression. While initially postulated to act as an oncosuppressor in normal cells, dysregulation of autophagy is now regarded as a progression factor for established cancer cells, which may, thus, gain drug resistance. Although the involvement of autophagy in the onset of drug resistance in ER^+^ breast cancer is now well recognized, the underlying molecular mechanisms are only marginally understood. Several reports point out that Tam and other SERMs and SERDs stimulate autophagy in ER^+^ breast cancer cells and tumors [6,7,8], and that expression of several autophagy-related proteins is often greater in metastatic tumors compared to primary breast cancer [9]. In keeping with this notion, pharmacological suppression of autophagy reverts drug resistance of breast cancer, either increasing or restoring susceptibility to the drugs [9,10]. For the reasons above, autophagy has been regarded as both a prosurvival mechanism counteracting the lethal effects of anticancer drugs and an attractive target for therapy of metastatic or chemotherapy-resistant breast cancer [11].

The lysosomal compartment is a key element of the autophagic machinery, accounting for the exhaustive degradation of the engulfed substrates [12]. First characterized as the main degrading organelles, lysosomes are now granted with complex regulatory functions in both normal and pathological conditions, including cancer development and progression [13]. Lysosomes are known to account for the lower susceptibility of cancer cells to chemotherapeutics which are weak bases by favoring their intralysosomal sequestration and ensuing removal from the cell by exocytosis [14,15]. However, intralysosomal accumulation of drugs to high concentrations, when not followed by their extracellular release, has also been reported to trigger lysosomal membrane permeabilization (LMP) and activation of lysosomal death pathways [16]. Thus, in this view, lysosomal destabilization may represent both a putative intrinsic mechanism of action of these drugs and an effective strategy to increase susceptibility of cancer cells to anticancer therapy [17,18,19]. In a recent report, Hultsh et al. found that lysosomes of a Tam-resistant subclone originated from T47D cells, a luminal A breast cancer cell line, are more abundant and bigger than those of parental cells [20]. Of interest, Tam-resistant T47D cells were more resistant to LMP induced by lysosomotropic agents than the parental ones, further linking increased lysosomal stability to Tam resistance.

In a previous study, we demonstrated that activation of the autophagic flux determines the lysosomal relocation of the cytoplasmic metallothioneins (MT), a class of iron-binding proteins, and grants rat hepatoma cells with a greater resistance to TNF-mediated cytotoxicity [21]. We also showed that the enhanced intralysosomal accumulation of MT abrogates the TNF-mediated LMP, eventually protecting hepatoma cells against the lethal action of the cytokine.

In this research, we investigated the putative molecular mechanisms that account for increased resistance to endocrine therapy of ER^+^ breast cancer cells by focusing on the lysosomal compartment. Chronical exposure of MCF7 cells to Tam produced a cell line resistant to 5 µM Tam (MCF7-TamR), which displays an increased autophagic flux, overexpression of the cytoplasmic iron-binding proteins MT2A and ferritin heavy chain, and a greater resistance to Tam-induced lysosomal membrane permeabilization (LMP) compared to the parental cells. We show here that autophagy plays a critical role in Tam resistance of MCF7-TamR cells by safeguarding the lysosomes against Tam-mediated LMP.

## 2. Results

### 2.1. Generation and Characterization of a Tamoxifen-Resistant Subclone Derived from MCF7 Cells

To generate a reliable in vitro model of endocrine-resistant breast cancer cells, we mimicked its onset by culturing the MCF7 cells in the presence of increasing concentrations of Tam up to 5 µM for about 15 months. At the highest concentration used, Tam significantly slowed the growth of parental MCF7 (Figure 1a) and affected cell-cycle distribution; S and G2/M phases were progressively depleted of cells, which accumulated in G0/G1 or, after 6 days of treatment, in a hypodiploid population likely representing dying cells (Appendix A). At the end of the selection time, a subclone of MCF7 cells, designated MCF7-TamR, capable of growing unrestrained in the presence of 5 µM Tam, was obtained. However, we observed that, even in the absence of Tam, the growth rate of MCF7-TamR cells was slower than that of parental cells; in fact, after 6 days of culture, the maximum cell density reached by MCF7-TamR was about half of that achieved by parental cells (Figure 1a,b). Next, the IC_50_ for Tam in the different cell lines was calculated after 48 h of continuous treatment with drug. According to the different growth pattern in the presence of the drug (Figure 1a,b) in MCF7-TamR cells, the calculated IC_50_ was 47.2 ± 7.42 µM, a value significantly greater than that of parental cells (12.25 ± 1.91 µM), but in good agreement with values previously reported by other authors [22,23]. To further assess the drug resistance of the subclone, we measured the growth of both susceptible and resistant cells in the presence of 12 µM Tam, representing the IC_50_ for parental MCF7 cells. As expected, the growth of MCF7 cells was completely blunted by the drug (Figure 1c). By contrast, growth of MCF7-TamR was effectively slowed by Tam in these experimental conditions, but not totally abrogated as in parental cells (Figure 1d).

Onset of endocrine resistance can be attributed to several mechanisms, the most common of which are the reduced expression or malfunctioning of the ER or upregulation of the PI3K/AKT/PTEN pathway. We first investigated the expression of both ER and pAkt in both cell lines and observed that neither the intracellular amount nor the Tam-dependent modulation of both ER and pAkt was significantly changed in MCF7-TamR versus MCF7 cells (Appendix A for the original blots). To verify whether endocrine resistance was afforded by the presence of a dysfunctional ER, we analyzed the estrogen response of the cells using the E-screen, a test originally devised to demonstrate the ER activation [24] and so far used to ascertain the estrogenic activity [25] of different substances. Proliferation of MCF7 cells was increased by 17-β estradiol in the range of 10–100 pM, which produced the greater proliferative response. As expected, 5 µM Tam abrogated the growth stimulation brought about by estradiol. MCF7-TamR cells responded to 17-β estradiol treatment in the range of 100 pM–10 nM, and again growth was strongly reduced by Tam, which suggests that ER of these cells retained the capability to respond to the estrogens and to be inhibited by Tam (Appendix A). Altogether, these results rule out the possibility that Tam resistance of this subclone depends on both a dysfunctional ER signaling and an upregulation of the PI3K/AKT/PTEN pathways.

### 2.2. Tamoxifen Affects the Lysosomal Compartment in MCF7 Cells

Since several anticancer drugs affect the lysosomal compartment of target cells [26,27], we next investigated whether Tam also had such an activity. Accordingly, 5 µM Tam triggered lysosomal membrane permeabilization (LMP), as evidenced by the appearance of cells with a weak and diffuse, rather than bright and punctate, LysoTracker Red fluorescence (Figure 2a, arrows); for such a reason, cells showing LMP were subsequently indicated as “pale cells”. Such cells became evident by day 2, i.e., earlier than growth reduction and cell death onset (Appendix A), and they increased to about 15% of the total population by days 4–6 of treatment (Figure 2b). In cells showing LMP, lysosome number/cell decreased with time (Figure 2c). In contrast, lysosome size tended to increase (Figure 2d), indicating that Tam markedly affected the lysosomal compartment of MCF7 cells early on. A reduction in SQSTM1/p62 and an accumulation of LC3-II were evident at days 2 and 4 of treatment (Figure 2e); abrogation of SQSTM1/p62 degradation and further accumulation of LC3-II in the presence of E64d and leupeptin confirmed that Tam triggered autophagic flux in MCF7 cells.

We next investigated whether the lysosomal effects of Tam were dose-dependent and affected cell viability. We exposed MCF7 cells to increasing concentrations of Tam for 24 h and observed that the number of cells undergoing LMP increased with Tam concentration (Figure 3a). Tam treatment dose-dependently affected viability of MCF7 cells (Figure 3b), but with a different kinetic from LMP development, which always preceded death. The reduction in viability, in fact, was negligible for Tam concentrations up to 15 µM, which, on the other hand, already triggered significant LMP. Cell death became highly significant only upon treatment with 20 µM Tam, which reduced the culture viability by about 50%. According to the above observation, clonogenic efficiency dropped dose-dependently and fell to about 50% with 20 µM Tam (Figure 3c). By considering the overall trend of both loss of viability and clonogenic efficiency, 20 µM can be regarded as the LD_50_ of Tam for MCF7 cells under our experimental conditions. Using this Tam concentration, we eventually observed that LMP appeared as early as after 20 min of treatment and occurred in about 90% of cells after 2 h (Figure 3d,e, first and second panels from left). In addition to increasing the percentage of pale cells, Tam treatment also brought about the release of cathepsin B from permeabilized lysosomes (Figure 3e, third and fourth panels from left). On the basis of these findings, and according to the fact that this treatment did not elicit significant cell death (Appendix A), we decided to use 20 µM Tam for most of the subsequent experiments.

### 2.3. Tam-Induced Alterations of the Lysosomal Compartment Are Reversible and Associated with Autophagy Activation

We next assessed whether Tam-induced LMP is a transient alteration in MCF7 cells. We used washout experiments in which cells were treated with 20 µM Tam for 2 h and then shifted to Tam-free medium for up to 24 h before viability assessment. We found that viability was significantly altered neither at the end of the 2 h of Tam treatment nor at the end of the 24 h recovery in normal growth medium after Tam removal (Figure 4a), and that this did not depend on the absence of LMP (Figure 4b). Rather, although massively induced by Tam treatment, LMP reverted to control levels as soon as 4 h after drug removal (Figure 4b). The number of lysosomes/cell and the overall lysosomal area (Figure 4c,d, respectively) followed the same trend and returned to control values within about 4 h of recovery. These findings suggest that Tam-induced lysosomal damage is reversible and that MCF7 cells may escape from death by attenuating Tam-induced LMP.

Since autophagy is considered to be essential for cell survival, we investigated whether autophagy or lysophagy was activated during recovery from Tam-induced LMP using the tandem fluorescent LC3 (tfLC3) [28,29] or galectin-3 (tfGal3) [30,31] reporters. We found that Tam treatment increased both autophagic flux and lysophagy (Figure 4e,f, respectively). The results show that both autophagy and lysophagy started early during Tam treatment and remained elevated during recovery after drug washout.

To probe the contribution of autophagy to MCF7 survival after Tam-induced lysosomal damage, MCF7 cells were treated with 20 µM Tam for 2 h in the presence of 3-methyladenine (3-MA) or chloroquine (CQ), which were kept with the cells during the 24 h of recovery after Tam washout. As expected, inhibition of autophagy during recovery after Tam treatment significantly reduced cell survival (Figure 4g), confirming that autophagy played a prosurvival role in Tam-treated MCF7 cells. The data gathered so far support the view that autophagic disposal of damaged lysosomes effectively counteracts the detrimental effects of Tam-induced LMP.

### 2.4. Tam-Resistant MCF7 Cells Are Less Prone to Undergo Tam-Induced Lysosomal Damage

We subsequently asked whether Tam resistance correlates with a greater capacity of cancer cells to resist drug-induced LMP. For these experiments, we used the MCF7-TamR cell line (Figure 1b,d). The lysosomes of MCF7-TamR cells displayed higher resistance to LMP induced by 20 µM Tam for 2 h than lysosomes of parental cells (Figure 5a–d). We then investigated whether the atypical resistance to Tam-induced LMP relies on the activation of autophagic flux and the impact of autophagy inhibition on drug susceptibility of MCF7-TamR cells. Consistent with the data above, we found that autophagic flux was significantly higher in MCF7-TamR than in parental cells (Figure 5e) and that impairing autophagy with a number of validated selective inhibitors restored susceptibility of resistant cells to 5 and 20 µM Tam (Figure 5f). In fact, while they did not significantly affect cell viability when used alone, the inhibitors reduced the viability of MCF7-TamR cells in the presence of Tam at both 5 and 20 µM. This finding demonstrates that restraining autophagy restored susceptibility to Tam of otherwise Tam-resistant breast cancer cells. Of interest, the Tam-sensitizing effect of autophagy inhibition, particularly that afforded by CQ and 3-MA, was evident already with 5 µM Tam, a concentration to which MCF7-TamR cells are resistant, and, as expected, more pronounced with 20 µM. Collectively, these results point out that Tam resistance of MCF7-TamR largely relies on autophagy, thus representing a survival mechanism.

### 2.5. Expression of Lysosome-Protecting Factors Is Dysregulated in MCF7-TamR Cells and Contributes to Tam Resistance

We then measured the relative expression of a set of genes linked to endocrine resistance in both MCF7 and MCF7-TamR cells (Table 1).

We first observed that the expression of ERα and β was almost unchanged between parental and resistant cells, a finding that ruled out that Tam resistance depended on the absence of ER. Autophagy-related protein 5 (Atg5) expression was unchanged, while that of Atg7 was moderately increased, suggesting that autophagy was activated. We then extended the analysis to genes belonging to the group of iron-binding proteins, which are known to protect the lysosomal compartment against drug-induced LMP [21,32,33] and already reported to be dysregulated in some breast cancer subtypes [34,35,36]. We found that both metallothionein 2A (MT2A) and ferritin heavy chain (FtH) mRNAs were significantly overexpressed in MCF7-TamR, while ferritin light chain (FtL) and heat-shock protein 70 (Hsp70) mRNA levels were essentially unchanged. Our findings demonstrate that dysregulation of autophagy and overexpression of iron-binding proteins coexist in MCF7-TamR cells.

Next, we investigated whether overexpression of the above iron-binding proteins is actually critical in conferring Tam resistance to MCF7-TamR cells. To this aim, we knocked down the iron-binding proteins significantly upregulated in this subclone and measured the proportion of cells undergoing LMP after treatment with 20 µM Tam for 2 h. Downregulation of MT, FtH, or Hsp70 with different small interfering RNAs (siRNAs) (Figure 5g) significantly increased the number of cells undergoing LMP. As observed in particular with the MT2A-2, FtH-2, and Hsp70-2 siRNAs, the level of LMP attained was similar to that of MCF7 cells treated with Tam in the same way (Figure 5h). This finding reveals that both overexpression of iron-binding proteins and elevation of autophagic flux contribute to the atypical Tam resistance displayed by MCF7-TamR cells.

### 2.6. Autophagy Grants Tam Resistance to MCF7 Cells by Protecting the Lysosomes from Drug-induced LMP

Next, we investigated whether dysregulated expression of the genes mentioned above protected the lysosomes from Tam-induced LMP and whether resistance to LMP correlated with Tam resistance. First, wildtype MCF7 cells were transiently transfected with plasmids encoding GFP alone, MT2A-GFP, FtH-GFP, or Hsp70-GFP (Appendix A for the original blots) and exposed to 20 µM Tam for 2 h before LMP assessment. Ectopic expression of all proteins, excluding GFP alone, protected lysosomes of parental cells from LMP (Figure 6a, microscopic pictures). Interestingly, LysoTracker Red-fluorescent lysosomes of transfected cells were also positive for GFP fluorescence of chimeric proteins (Figure 6a, graphs). This observation demonstrates that ectopic GFP-tagged proteins were transported to the lysosomes, presumably because of their autophagy-mediated sequestration and lysosomal delivery. In keeping with the morphological analysis, LMP dramatically decreased in transfected compared to non-transfected and GFP-transfected MCF7 cells (Figure 6b), confirming that autophagy-mediated lysosomal accumulation of iron-binding proteins prevented Tam-induced lysosomal damage even when cells were exposed to high concentrations of the drug.

Next, we verified whether overexpression of iron-binding proteins and their autophagy-mediated contribution to restrain Tam-induced LMP allowed higher survival after Tam treatment. We performed a clonogenic assay using MCF7 cells stably transfected with GFP alone, MT2A-GFP, or Hsp70-GFP. Tam strongly reduced clonogenic capability of both non-transfected and GFP-transfected cells, but not that of cells overexpressing either MT2A or Hsp70 (Figure 6c). Of interest, although the number of colonies formed by Tam-treated, MT2A-transfected, or Hsp70-transfected MCF7 cells was similar to that of untreated cells, their size was smaller (data not shown), which was consistent with the reduced growth rate of Tam-resistant cells compared to parental ones (as evidenced by Figure 1a,b, respectively).

To better clarify whether prevention of LMP produced by overexpression of iron-binding proteins depends on autophagy, we assessed lysosomal integrity in MCF7 cells transiently transfected as above in the presence of the autophagy inhibitor 3-MA. Examination of lysosomes by confocal microscopy confirmed that overexpression of MT2A, FtH, or Hsp70 in the absence of 3-MA prevented LMP induction by 20 µM Tam (Figure 7, third row). Conversely, inhibition of autophagy completely abrogated lysosomal protection as evidenced by the greatly reduced number of LysoTracker Red- and GFP-positive lysosomes, indicating that lysosomal integrity depended on functioning autophagy (Figure 7, fourth row).

Collectively, the present results support our hypothesis that autophagy-mediated lysosomal delivery of protective factors triggers Tam resistance of MCF7-TamR cells by reducing Tam-induced LMP and cytotoxicity.

### 2.7. Overexpression of Iron-Binding Proteins and Activation of Autophagy Are Required to Restrain Tam-Mediated LMP in Breast Cancer Cell Lines of the Luminal A Subtype

Eventually, we checked whether the above conclusions gathered from MCF7 cells extended to other breast cancer cell lines by investigating the susceptibility to Tam-induced LMP of other cell lines of the luminal A subtype such as T47D, MDA-MB-415, and ZR-75-1 [37]. The lysosomal compartment of MDA-MB-415 and ZR-75-1 cells was heavily affected by treatment with 20 µM Tam for 2 h, a finding that was in good agreement with the results gathered with MCF7 cells (Figure 8a). Furthermore, the growth of both MDA-MB-415 and ZR-75-1 was significantly slowed starting from 3 days of culture in the constant presence of 5 µM Tam (Figure 8b). By contrast and quite surprisingly, the lysosomes of T47D cells were only marginally affected by Tam treatment, in strict analogy to that observed in MCF7-TamR cells (Figure 8a).

Since an increased intracellular level of iron-binding proteins accounts for Tam-resistance of MCF7-TamR cells, we also investigated the relative expression of these lysosome-protecting factors in all the other cell lines (Figure 8c). The analysis revealed that T47D cells overexpress both Hsp70 and MT2A, which could explain their high resistance to Tam-induced LMP. In order to verify this possibility, we silenced either Hsp70 or MT2A in T47D cells before measuring LMP. As expected, silencing significantly increased susceptibility of these cells to Tam-induced LMP (Figure 8d). This finding agrees with the results obtained in the similar experiment performed with MCF7-TamR cells (see Figure 5h).

Figure 8c also shows that MDA-MB-415 and ZR-75-1 cells overexpress some iron-binding proteins (Hsp70, FtH, and MT2A, or Hsp70, respectively); however, in spite of this, both retained a high susceptibility to Tam-induced LMP (Figure 8a), which apparently argues against the postulated lysosome-protecting activity of these proteins. To rule out the possibility that overexpressed endogenous iron-binding proteins might be mutated or dysfunctional in these cell lines and, thus, unable to prevent Tam-induced LMP, MDA-MB-415 and ZR-75-1 cells were transiently transfected with plasmids encoding GFP, MT2A-GFP, FtH-GFP, or Hsp70-GFP (Appendix A) and subsequently treated with 20 µM Tam for 2 h. However, lysosome detection and counting by confocal microscopy revealed that enforced expression of the iron-binding protein-GFP did not further protect the lysosomes against Tam-induced damage (Figure 9a,b, and Appendix A, respectively). We next verified whether such an unexpected lack of lysosome protection depended on an impaired Tam-induced activation of autophagy, as otherwise observed in MCF7 and T47D cells. Tam treatment triggered LC3-II accumulation in both MDA-MB-415 and ZR-75-1 cells (Figure 9b,d); however, its amount was not significantly increased by E64d and leupeptin, which suggests that Tam does not stimulate autophagic flux in these cells. To test whether activation of autophagy restrains Tam-induced lysosomal damage, transfected cells were starved for 2 h in HBSS before evaluation of lysosome integrity. Starvation almost completely protected the lysosomal compartment of both cell lines from Tam-mediated damage and restored the lysosome number to the values of untreated cells (Figure 9a,c; Appendix A). These findings definitely demonstrate that both dysregulation of lysosome-protecting factors and activation of autophagy are required to enhance the resistance of breast cancer cells to Tam-induced lysosomal damage.

## 3. Discussion

Intrinsic or acquired endocrine resistance is a major clinical complication that worsens the prognosis for about 30% of breast cancer patients [38]. During the last few years, autophagy has been recognized as a key player behind drug resistance of breast cancer cells and, consequently, identified as a potential target for circumventing it [39,40]. However, overcoming drug resistance by targeting autophagy still falls short of being achieved, mostly because there is no clear understanding of the subtle molecular mechanisms via which autophagy drives drug resistance.

In this research, we show that Tam triggers both lysosomal damage and autophagy in MCF7 cells. The finding that LMP also occurs with Tam concentrations that can be achieved in breast cancer tissue in vivo [41,42] suggests that Tam-mediated lysosomal damage may be a significant feature of anticancer activity of Tam. Lysosomes have been reported to play a critical role in both triggering cancer cell death and inducing drug resistance via different mechanisms. Several commonly used anticancer agents induce LMP and lysosomal death, confirming that these organelles are suitable therapeutic targets for cancers [43,44]. By contrast, lysosomal sequestration and subsequent exocytosis of anticancer agents that are weak hydrophobic bases limit the intracellular availability of active drug(s) and contribute to drug resistance [14,45,46]. In our system, LMP induced by a 2 h treatment with 20 µM Tam did not cause sudden death, which showed up only when autophagy was impaired. This evidence further highlights autophagy as a survival response of Tam-treated breast cancer cells [47,48] and supports the possibility that its activation reduces Tam toxicity by antagonizing drug-induced lysosomal damage. Activation of autophagy might theoretically lead to survival either by promoting lysophagy, the selective removal of Tam-damaged lysosomes [31,49], or by preventing lysosomal damage itself through alternative autophagy-based protective mechanisms. The first mechanism seems to operate in MCF7 cells, in which lysophagy is promptly activated in response to a 2 h exposure to Tam to remove dysfunctional lysosomes that might trigger cell death. By contrast, in cells transiently overexpressing the known regulators of intracellular iron availability MT2A, FtH, or Hsp70 [32,50,51], an alternative protective mechanism based on autophagic sequestration and lysosomal delivery of these proteins prevents Tam-induced lysosomal damage and promotes cell survival. The models used in our study seem to confirm such a dual possibility, since lysosomes of transiently transfected MCF7 accumulated iron-binding proteins and became more resistant to Tam-induced damage compared to non-transfected cells. In these conditions, pharmacological inhibition of autophagy completely abrogated lysosome protection, confirming its critical role in cancer cell survival. The same kind of response was observed in MCF7-TamR cells, which overexpress MT2A and FtH as a result of adaptation to Tam. Interestingly, Tam-resistant cells, which also show a higher autophagic flux and a lower proneness to undergo Tam-induced LMP than parental cells, were re-sensitized to therapeutic concentrations of Tam when either autophagy was impaired or the overexpressed lysosome-protecting factors were downregulated. The present and previous observations from our group [21] demonstrate that overexpression of lysosome-protecting factors is inherently associated with drug resistance of cancer cells. They also suggest that autophagy contributes to drug resistance far beyond the production of anabolic units to support the metabolic requirements of cancer cells facing various kinds of stress. Our findings gathered from both MCF7-TamR and other breast cancer cell lines of the luminal A molecular subtype agree with clinical observations that overexpression of MT2A, FtH, or Hsp70 represents a poor prognostic factor for metastatic breast cancer of different subtypes, which significantly correlates with their aggressiveness and drug resistance [35,36,52]. Our results further support the possibility that upregulation of factors safeguarding lysosome integrity against LMP induced by a variety of agents [20,32,33,53,54] may contribute to drug resistance in breast cancer if autophagy is concurrently activated or dysregulated. This paradigm is evidenced by the information gathered from T47D, MDA-MB-415, and ZR-75-1 cells. Although all these lines are validated models of breast cancer of the luminal A molecular subtype, they show a different pattern of response to Tam treatment. Although T47D cells are known to express the ER, their growth has been reported to be less affected by Tam compared to MCF7 or other luminal A cells [55,56]. We show here that these cells overexpress Hsp70 and MT2A, and that they are intrinsically highly resistant to Tam-induced LMP. We also provide a demonstration that knockdown of these proteins restores susceptibility of these cells to LMP. By contrast, MDA-MB-415 and, even if to a lesser extent, ZR-75-1 cells have been found to overexpress most of the lysosome-protecting factors investigated in this research; however, in spite of this finding, both appear to be as susceptible to LMP as MCF7 cells. In particular, in these two cell lines, we also detected a lower capability of Tam to trigger the autophagic flux compared to what occurs in MCF7 and T47D cells. Of interest, in these cells, autophagy inhibitors revealed that Tam leads to a blockade of autophagy, rather than its activation. This, by impairing the autophagy-mediated delivery of these proteins to the lysosomes, forestalls the protection of lysosomal compartment against Tam-induced damage. Our results might impact the choice of the best therapeutic approach for advanced, endocrine-resistant ER^+^ breast cancer with an upregulated PI3K/PTEN/Akt/mTOR pathway [57]. In such cases, endocrine inhibitors are routinely combined with rapamycin analogues (“rapalogues”) that inhibit the mechanistic target of rapamycin complex 1 [1,58,59]. Rapalogues are clinically useful for several cancers, including breast cancer, but they are also powerful inducers of autophagy [60,61,62]. According to our present observation with MDA-MB-415 and ZR-75-1 cells, in endocrine-resistant breast cancers overexpressing MT2A, FtH, or Hsp70, rapalogues might enhance the autophagy-mediated relocation of these factors to the lysosomes, blunting LMP and further exacerbating drug-resistance. A set of experimental investigations and preclinical studies indeed revealed that pharmacological inhibition of autophagy increases the clinical effectiveness of rapalogues [63].

The findings gathered by our investigation demonstrate that autophagy-mediated suppression of Tam-induced lysosomal damage is a likely mechanism via which autophagy may trigger endocrine resistance in MCF7 cells that overexpress iron-binding proteins or other lysosome-protecting factors. Correspondingly, screening for overexpression of these factors might lead to early identification of ER^+^ breast cancer patients at risk for endocrine resistance and help to devise the most suitable therapeutic approaches to minimize the risk and optimize the effectiveness of anticancer treatment.

## 4. Materials and Methods

### 4.1. Cell Cultures

MCF7 cells were grown in DMEM (D6429, Merck, Milan, Italy) supplemented with 10% fetal bovine serum, 100 U/mL penicillin, and 100 µg/mL streptomycin. For experiments, cells were seeded at 1.5 or 3 × 10^4^/cm^2^ and, 24 h later, exposed to Tam (sc-208414, Santa Cruz Biotechnology, Heidelberg, Germany) in complete growth medium; for long-term treatments, Tam was replaced every third day. Cell line authentication was performed at BMR Genomics. T47D cells were kindly provided by Prof. M. De Bortoli and Prof. F. Cavallo and maintained in RPMI 1640 or DMEM, respectively, with supplements as indicated for MCF7 cells. MDA-MB-415 and ZR-75-1 cells were obtained from Cell Bank, Interlab Cell Line Collection, Genova, Italy and routinely grown as described for MCF7 cells; for ZR-75-1 cells, 1% nonessential amino acids were added to the culture medium.

The Tam-resistant subclone (MCF7-TamR) was generated by growing the parental MCF7 cells in DMEM with increasing concentrations of Tam. Starting from 0.1 µM Tam, the concentration of drug was approximately doubled (0.25, 0.5, 1, 2, 4, and 5 µM) every time the cells restarted to grow in the presence of the drug. Over a selection time of about 15 months, cells gained the capability to stably grow in the presence of 5 µM Tam. Following selection, the cells were maintained under constant presence of Tam in the culture medium. With the only exception being the data reported in Figure 4g and Appendix A, all the experiments performed with this subclone were made with cells maintained in the presence of 5 µM Tam.

For determination of the IC_50_, MCF7 and MCF7-TamR cells were seeded in 96-wells plates at 2 × 10^4^/cm^2^ in normal growth medium and allowed to adhere for 48 h as described [23]. Cells were then incubated in the presence of 0.01, 0.1, 1, 10, 100, or 1000 µM Tam for an additional 48 h, after which the relative cell number was measured by MTT as described for the viability assay. The IC_50_ for each cell line was calculated using either CalcuSyn (Biosoft, Cambridge, UK) or the freely available IC_50_ Calculator tool (AAT Bioquest, https://www.aatbio.com/tools/ic50-calculator (accessed date on 10 November 2020)).

### 4.2. Viability Assay

Viability was evaluated with the MTT test. Cells were seeded in 96-well plates at 1.5 × 10^4^ cells/cm^2^ in 100 µL of growth medium; after treatments, 20 µL of a 5 mg/mL solution of MTT (M2128, 3-(4,5-dimethyl-2-thiazolyl)-2,5-diphenyl-2*H*-tetrazolium bromide, Sigma-Aldrich, Milan, Italy) was added to each well for 2 h at 37 °C under normal growth conditions. The formazan precipitates that form by activity of the mitochondrial dehydrogenases in living cells were dissolved in 100 µL of DMSO by rocking the plates over an orbital shaker for 60 min at room temperature before measurement of the absorbance at 595 nm with the iMark Microplate Reader (Bio-Rad Laboratories, Segrate, Italy).

### 4.3. Flow Cytometry

For cell-cycle analysis, cells were detached with trypsin, centrifuged at 600× *g* for 10 min at 4 °C, resuspended in cold 70% ethanol for 30 min, washed, and resuspended in PBS. After staining with propidium iodide (180 µg/mL), aliquots of 5000 cells/sample were analyzed with an Accuri C6 flow cytometer (Becton-Dickinson, Milan, Italy).

For cell death assessment, cells treated or not with 5 or 20 µM Tam for 48 or 2 h, respectively, were detached and centrifuged as above; then, they were resuspended in 200 µL of binding buffer before addition of 5 µL annexin-V–FITC (BMS500FI/100, Annexin V–FITC Apoptosis detection kit, Affymetrics, eBioscience, Bender MedSystems GmbH, Vienna, Austria). After 10 min of incubation at room temperature in the dark, 10 µL of a 20 µg/mL propidium iodide solution was added; at least 5000 cells/sample were analyzed with an Accuri C6 flow cytometer.

### 4.4. Analysis of Lysosomal Alterations

#### 4.4.1. Lysosomal Membrane Permeabilization

For LMP analysis, cells were loaded with 100 nM LysoTracker Red DND-99 (L7528, ThermoFisher Scientific, Monza, Italy) for 20 min at the end of Tam treatments. After dye removal, cells were imaged in phenol red-free DMEM with a Leica or Axiovert-35 epifluorescence or an LSM800 confocal microscope (Zeiss, Milan, Italy). Cells displaying either less intensely red-stained lysosomes or a cytoplasmic diffuse red fluorescence were cells undergoing LMP and were routinely indicated “pale cells”.

Release of cathepsin B-GFP (CB-GFP) was assessed in cells transfected as described [64] with the pCB-GFP (a gift of G. Gores, Mayo Clinic, College of Medicine, Rochester, MN, USA) using the K2 Transfection system (T060-0.75, Biontex Laboratories GmbH, München, Germany). After treatments, cells were fixed with 4% paraformaldehyde for confocal examination.

#### 4.4.2. Lysosome Number, Lysosomal Area, and Mean Lysosomal Fluorescence

The number of lysosomes/cell, their mean surface, and their relative fluorescence were measured in cells stained with LysoTracker Red as above. Images acquired with the fluorescence or confocal microscope were analyzed with the Spot detector module of the Icy analytical software, version 2.1.0.1, BioImage Analysis unit, Institut Pasteur, Paris, France [65].

### 4.5. Clonogenic Assay

MCF7 cells were seeded at 600 or 100 (for Figure 2c and Figure 5c, respectively) cells/well in 24-well plates, allowed to adhere overnight, and treated for 24 h with the indicated concentrations of Tam. The medium was then replaced and survivors allowed to grow for 14 days, after which the established colonies were fixed with 4% paraformaldehyde, stained for 20 min with 0.1% (*w*/*v*) crystal violet, washed with distilled water, and counted with the Colony Counter macro of ImageJ. Only for the clonogenic assay presented in Figure 5c, cells were transfected with plasmids encoding GFP, MT2A-GFP, or Hsp70-GFP. Stable transfectants were selected with 700 µg/mL G418 (A1720-1G, Sigma Aldrich) and used for experiments. Expression of the relevant proteins was confirmed by fluorescence microscopy examination of the monolayers.

### 4.6. Real-Time RT-PCR

Total RNA was extracted with the TriReagent (T9424, Sigma Aldrich,) and reverse-transcribed with the FireScript RT cDNA synthesis Kit (06-12-00200, Solis BioDyne, Tartu, Estonia). Next, 50–100 ng RNA/sample was amplified in a CFX Connect (Bio-Rad Laboratories,) with the HOT FIREPol Evagreen qPCR Supermix (08-36-00001, Solis BioDyne). PCR primers (Appendix A) were designed with Primer3. The relative mRNA level was calculated using the 2^−ΔΔCT^ method (Bio-Rad Maestro, Bio-Rad Laboratories).

### 4.7. Overexpression and Analysis of Lysosomal Relocation of Iron-Binding Proteins

MCF7 cells were transiently transfected 24 h after seeding in µ-Slide 8 well (80826, Ibidi GmbH, Gräfelfing, Germany) with plasmids encoding human MT2A-GFP (Addgene plasmid 11613, a gift from Steven Johnson) and Hsp70-GFP (Addgene plasmid 15215, a gift from Lois Greene). The FtH-GFP vector was constructed by subcloning the complementary DNA (cDNA) of human wildtype Ft heavy chain, excised from the pUD-HFt plasmid (a gift from Sonia Levi) [66], into the pEGFP-N2. Then, 48 h after transfection, cells were treated with Tam in the absence or presence of 3-methyladenine (3-MA), stained with LysoTracker Red DND-99, and analyzed with a confocal microscope. Green and red fluorescence was measured using the Intensity Profile module of the Icy analytical software, version 2.1.0.1, BioImage Analysis unit, Institut Pasteur, Paris, France.

### 4.8. Western Blotting

Analysis of p62/SQSTM1, LC3 (P0067 and L7543, Sigma-Aldrich), ERα, GFP (sc-8005 and sc-9996, Santa Cruz Biotechnology), pAkt, and Akt (9271S and 4691, respectively, Cell Signaling, Danvers, MA, USA), and normalization versus β-actin (A5441, Sigma-Aldrich) were performed as described [21]. Bands were detected with the ChemiDoc XRS+ Imaging System (Bio-Rad Laboratories).

### 4.9. Determination of Autophagic Flux and Lysophagy

A total of 3 × 10^4^ cells/cm^2^ were seeded onto a µ-Slide 8 well (80826, Ibidi GmbH) and transfected with either ptfLC3 or ptfGal3 (Addgene plasmids 21074 and 64149, respectively) for measurement of autophagic flux and lysophagy. For washout experiments, cells transfected with the indicated plasmids were exposed 24 h later to 20 µM Tam for 2 h and allowed to recover for up to 24 h in normal growth medium. Following fixation with 4% paraformaldehyde, samples were imaged with the LSM800 confocal microscope. The number of yellow- and red-fluorescent dots representing the autophagosomes and autolysosomes or the lysosome-containing autophagosomes and autolysosomes in ptfLC3- or ptfGal3-transfected cells, respectively, was counted with the Green and Red Puncta Colocalization ImageJ Plugin (R.K. Dagda (University of Nevada School of Medicine), D. Shiwarski (Carnegie Mellon University) and C.T. Chu (University of Pittsburgh)) on a number of cells indicated in the relevant bars, extracted from at least three different microscopic fields/condition.

### 4.10. siRNA-Mediated Downregulation of Iron-Binding Proteins

Silencing of FtH, Hsp70, and MT2A was achieved by transfecting the cells with the Silencer Select predesigned siRNAs (Ambion, Carlsbad, CA, USA). For each target, two siRNAs were used: for FtH, siRNAs s5383 and s5385; for Hsp70, siRNAs s194536 and s6965; for MT2A, siRNAs s194629 and s226631. siRNAs were transfected at the final concentration of 20 nM with the Lipofectamine RNAiMAX (13778100, ThermoFischer Scientific) for 72 h, after which the cells were treated with Tam as indicated; silencing was verified by real-time RT-PCR. As a control, the Silencer Select Negative Control n° 1 siRNA (Ambion) was used at the same concentrations and for the same times of the other siRNAs.

### 4.11. Statistical Analysis

Data represent the mean ± SD of at least three independent experiments, each performed in triplicate. Differences between groups were analyzed with the one-way ANOVA followed by a Student–Newman–Keuls post hoc test or by the Student’s *t*-test using the Instat package (Version 3.10, GraphPad Software, San Diego, CA, USA). A *p*-value < 0.05 was considered statistically significant.

## 5. Conclusions

Autophagy plays a relevant role in the onset of endocrine resistance of ER^+^ breast cancer. Although several hypotheses have been proposed, a unifying theory explaining how this occurs at the molecular level is still lacking. Here, we show that Tam damages the lysosomal compartment and triggers LMP in all the ER^+^ breast cancer cell lines tested. Collectively, our data, thus, demonstrated that LMP may represent an important mechanism via which this anticancer drug exerts its biological action. At the same time, our findings revealed that susceptibility to Tam-induced LMP varies according both to the capability of Tam to activate autophagy in target cells and to the level of cytoplasmic iron-binding proteins. In fact, T47D cells, which overexpress MT2A and Hsp70, are less prone to undergo Tam-induced LMP and cell death compared to MCF7 and the other luminal A cell lines tested. On the other hand, silencing MT2A or Hsp70 in these cells restores LMP, which confirms that these iron-binding proteins act as lysosome-protecting factors that restrain drug-induced lysosomal damage and ensuing cytotoxicity.

However, our findings also revealed that the level of lysosome-protecting factors is not the unique factor that determines the proneness to Tam-induced LMP. MDA-MB-415 and ZR-75-1 cells, which overexpress MT2A or Hsp70 similarly to T47D cells, are susceptible to Tam-induced LMP comparably to MCF7 cells. Analysis of autophagic flux showed that, in these cells, Tam does not trigger autophagy, which prevents the lysosome-protecting effect brought about by MT2A or Hsp70 overexpression from showing up. In keeping with this result, we demonstrated that enforced activation of autophagic flux by starvation restores protection of the lysosomal compartment against Tam-induced LMP. Collectively, our data confirmed that the activation of autophagy, which occurs as a consequence of Tam treatment, contributes to Tam resistance of ER^+^ breast cancer cells by relocating inside the lysosomal lumen the protective factors capable of restraining the drug-induced lysosomal damage.

In conclusion, our results demonstrated that lysosome-protecting factors, such as the iron-binding proteins, by synergizing with activated autophagy, might represent an additional risk factor for onset of Tam resistance. According to this view, early identification of breast cancer patients which overexpress such protective factors might help to deploy suitable therapeutic strategies to limit the onset, as well as to overcome an already established drug resistance.

## Figures and Tables

**Figure 1 cancers-13-01252-f001:**
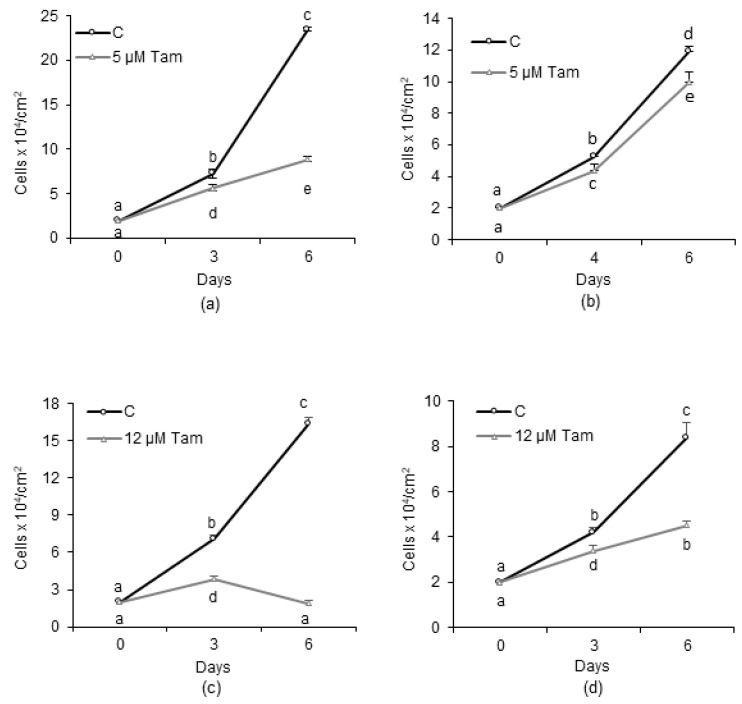
Effect of tamoxifen (Tam) on growth of human breast cancer cell lines. (**a**,**b**) Growth of MCF7 and MCF7-TamR cells, respectively, in the absence or presence of 5 µM Tam. (**c**) Growth of MCF7 and MCF7-TamR cells (**d**) in the presence of 12 µM Tam, which represents the calculated IC_50_ for Tam for the parental cell line. Data represent the mean ± SD of at least three independent experiments. Different letters indicate statistically different means as determined by ANOVA followed by a Student–Newman–Keuls post hoc test; statistical significance was set at *p* < 0.05.

**Figure 2 cancers-13-01252-f002:**
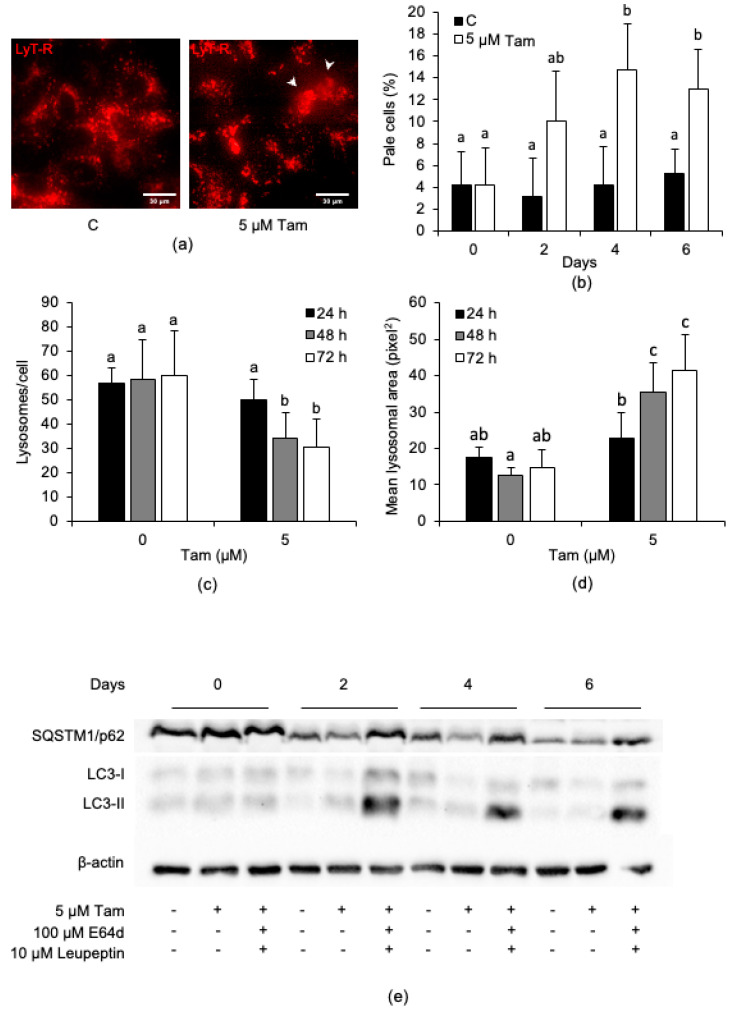
Tam (5 µM) triggers lysosomal membrane permeabilization (LMP) and autophagic flux in MCF7 cells. (**a**) LMP triggered in MCF7 cells (arrows) after 6 days of Tam treatment. (**b**) Number of pale cells, counted every second day of Tam treatment. (**c**) Effect of Tam on the number of lysosomes/cell and (**d**) on the mean lysosomal area. For each sample, at least 25 single cells were used for measurements. (**e**) Relative amount of the autophagy substrate SQSTM/p62 and of LC3-I/II analyzed for up to 6 days of continuous Tam treatment; refer to Appendix A for the original blots. The pictures in panel (**a**,**e**) are representative of three independent experiments. Data and statistical analysis are as in Figure 1.

**Figure 3 cancers-13-01252-f003:**
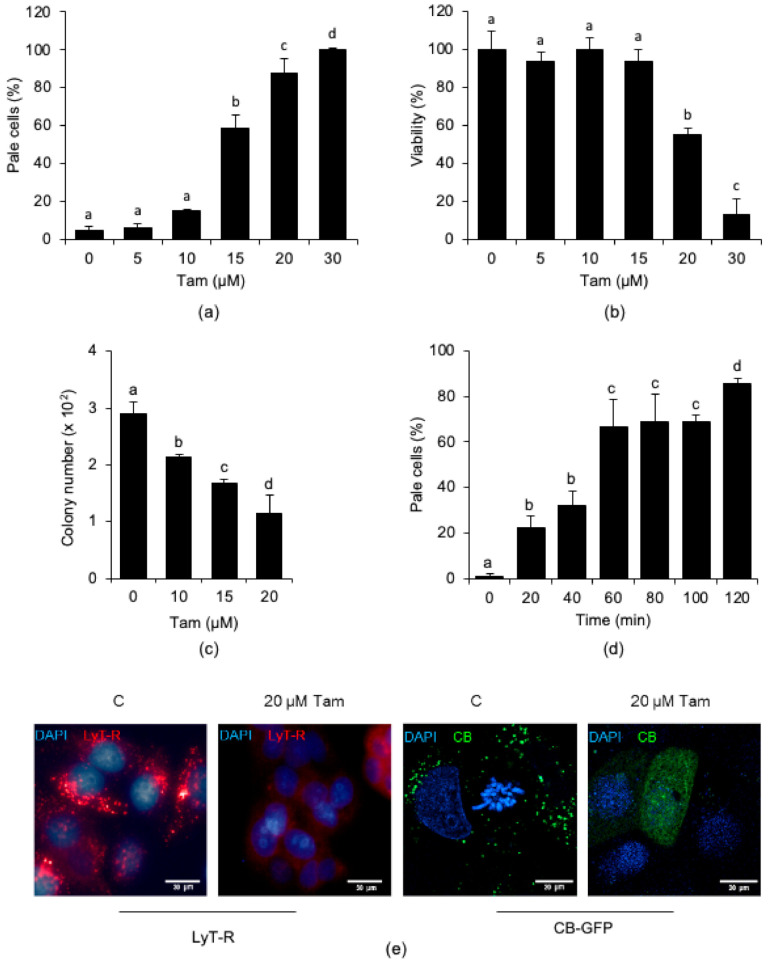
Tam-induced LMP is dose- and time-dependent in MCF7 cells. (**a**,**b**) Dose-dependency of Tam-induced LMP (**a**) or loss of viability (**b**) in cells treated for 24 h with Tam as indicated. (**c**) Reduction in clonogenic potential of MCF7 cells exposed for 24 h to the indicated concentrations of Tam. (**d**) Time-course of LMP induced by 20 µM Tam. (**e**) LMP (first and second panels) and release of cathepsin B-GFP (CB-GFP; third and fourth panels) induced by treatment with 20 µM Tam for 2 h; C and 20 µM Tam indicate controls and MCF7 cells treated with 20 µM Tam for 2 h, respectively. For panels (**a**–**d**), data and statistical analysis are as in Figure 1. Pictures in (**e**) are representative of three independent experiments.

**Figure 4 cancers-13-01252-f004:**
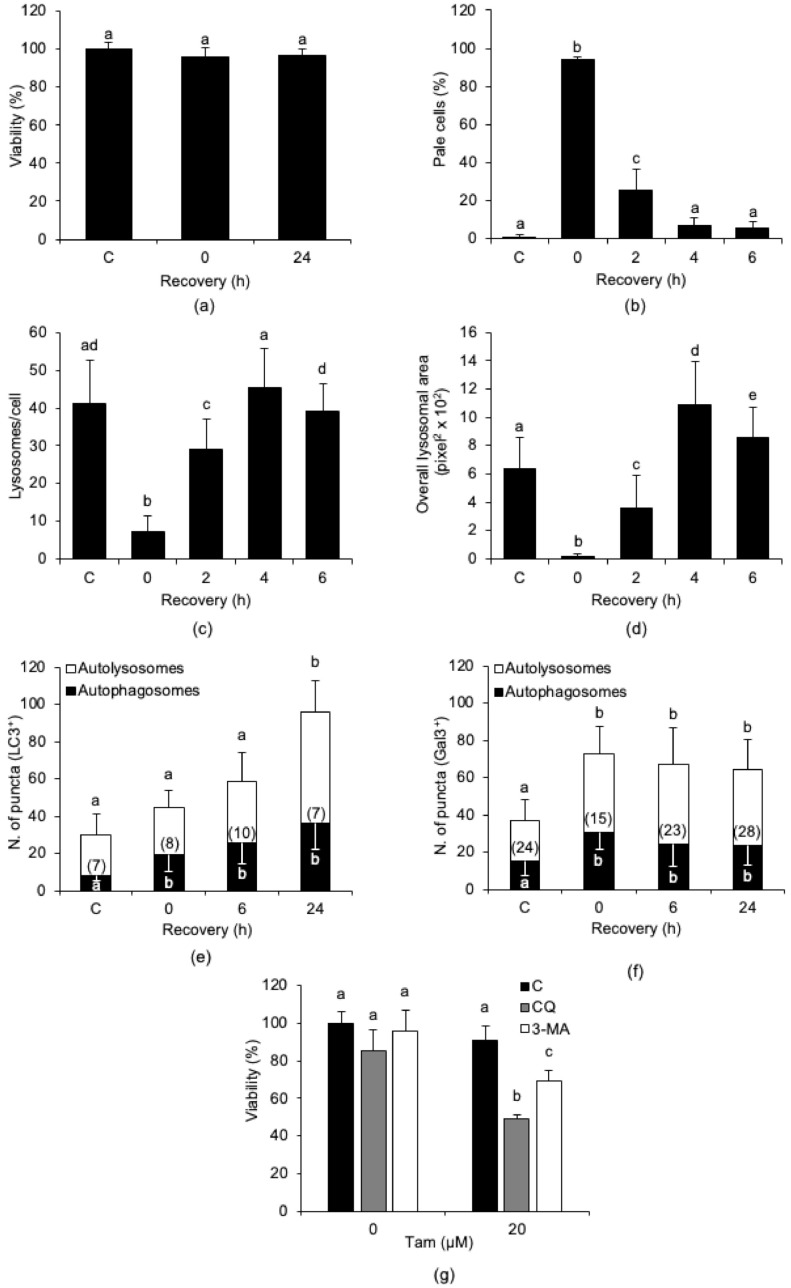
Tam-induced LMP is transient, and its reversion is accompanied by autophagy activation. (**a**,**b**) Effect of drug removal and of recovery in normal growth medium for up to 24 h on viability (**a**) and LMP induction (**b**) in MCF7 cells treated with 20 µM Tam for 2 h. Viability and LMP in controls were measured before Tam addition; Tam was removed at time 0. (**c**,**d**) Effect of Tam removal on lysosome number (**c**) and overall area of lysosomal compartment (**d**). (**e**,**f**) Activation of autophagy (**e**) and lysophagy (**f**) in MCF7 cells exposed to Tam and allowed to recover as in (**a**). The number of cells analyzed is shown in each bar. (**g**) Effect of autophagy inhibition on viability of MCF7 cells treated as in (**a**) and allowed to recover for 24 h. Data and statistical analysis are as in Figure 1. C: control MCF7 cells.

**Figure 5 cancers-13-01252-f005:**
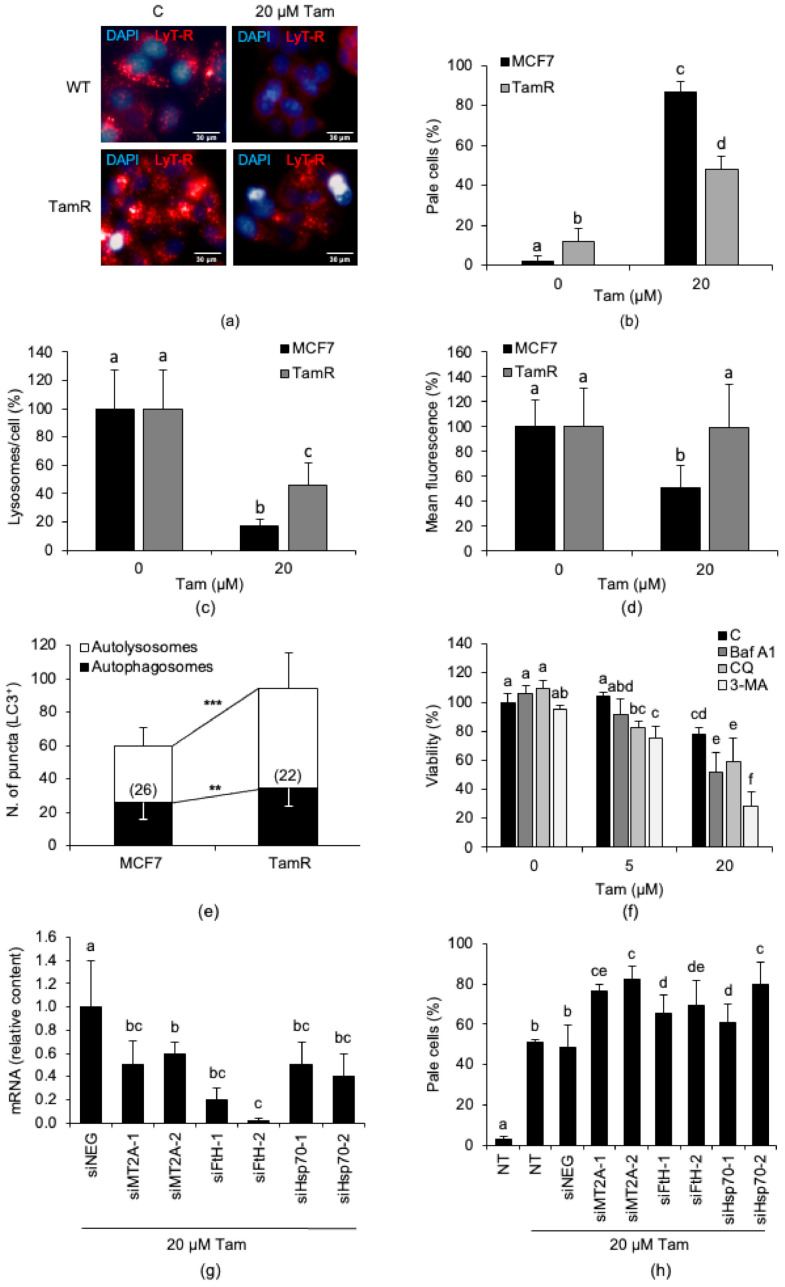
MCF7-TamR cells are resistant to Tam-induced growth inhibition and LMP and show a higher autophagic flux. (**a**) Effect of treatment with 20 µM Tam for 2 h on LMP in MCF7 and MCF7-TamR cells. (**b**) LMP induced by 20 µM Tam for 2 h on MCF7-TamR cells. (**c**,**d**) Effect of treatment with 20 µM Tam for 2 h on the number of lysosomes/cell (**c**) and the mean lysosomal fluorescence (**d**). (**e**) Autophagic flux in parental and MCF7-TamR cells. The number of cells analyzed is indicated in each bar. (**f**) Effect of autophagy inhibition with chloroquine (CQ, 50 µM), 3-methyladenine (3-MA, 5 mM), or bafilomycin A1 (Baf A1, 1 µM) on viability of MCF7-TamR cells exposed to 5 or 20 µM Tam for 24 h. CQ or 3-MA was added 60 min before Tam; Baf A1 and Tam were added simultaneously. (**g**) Relative content of MT2A, FtH, or Hsp70 messenger RNAs (mRNAs) in MCF7-TamR cells transfected for 72 h with 20 nM of the indicated small interfering RNAs (siRNAs). (**h**) LMP triggered by 20 µM Tam for 2 h in TamR cells silenced or not for 72 h with 20 nM of indicated siRNAs. WT: wildtype parental MCF7 cells; TamR: MCF7-TamR cells; C: untreated cells. NT: non-transfected cells; siNEG: control siRNA. Data and statistical significance of panels (**b**–**d**) and (**f**–**h**) are as in Figure 1; for panel (**e**), the Student’s *t*-test was used. ** *p* < 0.05 and *** *p* < 0.001. Panel (**b**) shows pictures representative of three experiments.

**Figure 6 cancers-13-01252-f006:**
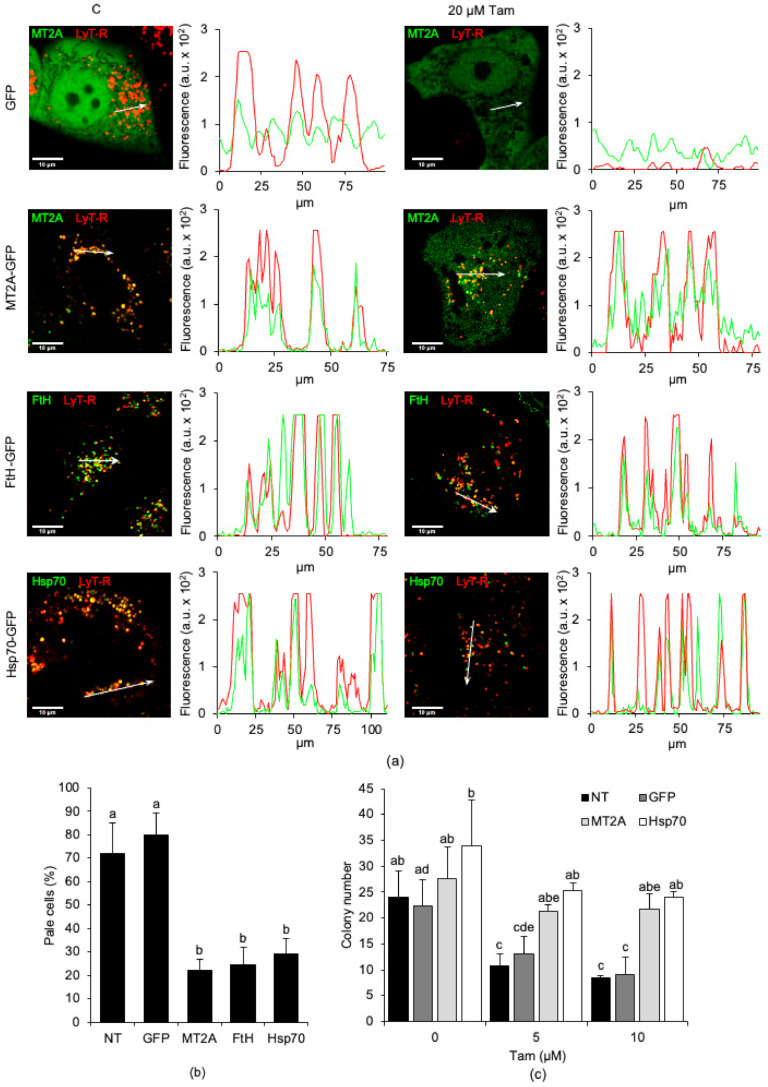
Overexpression and lysosomal relocation of MT2A, FtH, or Hsp70 abrogate Tam-induced LMP and elicit Tam resistance in MCF7 parental cells. (**a**) Lysosomal relocation of GFP, MT2A-GFP, FtH-GFP, or Hsp70-GFP (from first to fourth row, respectively). Images are representative of at least three independent transfections. Intensity plots (arbitrary intensity units) of red and green fluorescence of the organelles encompassed by the white arrow. (**b**) Attenuation of LMP induced by 20 µM Tam for 2 h in MCF7 parental cells non-transfected or overexpressing GFP, MT2A, FtH, or Hsp70. LMP was evaluated only in GFP-expressing cells, counted in three to five microscopic fields from three independent transfections. (**c**) Clonogenic potential of MCF7 cells non-transfected or stably transfected with GFP, MT2A, or Hsp70 and exposed to 5 or 10 µM Tam for 24 h. NT: non-transfected parental cells. For panels (**b**,**c**), data and statistical analysis are as in Figure 1.

**Figure 7 cancers-13-01252-f007:**
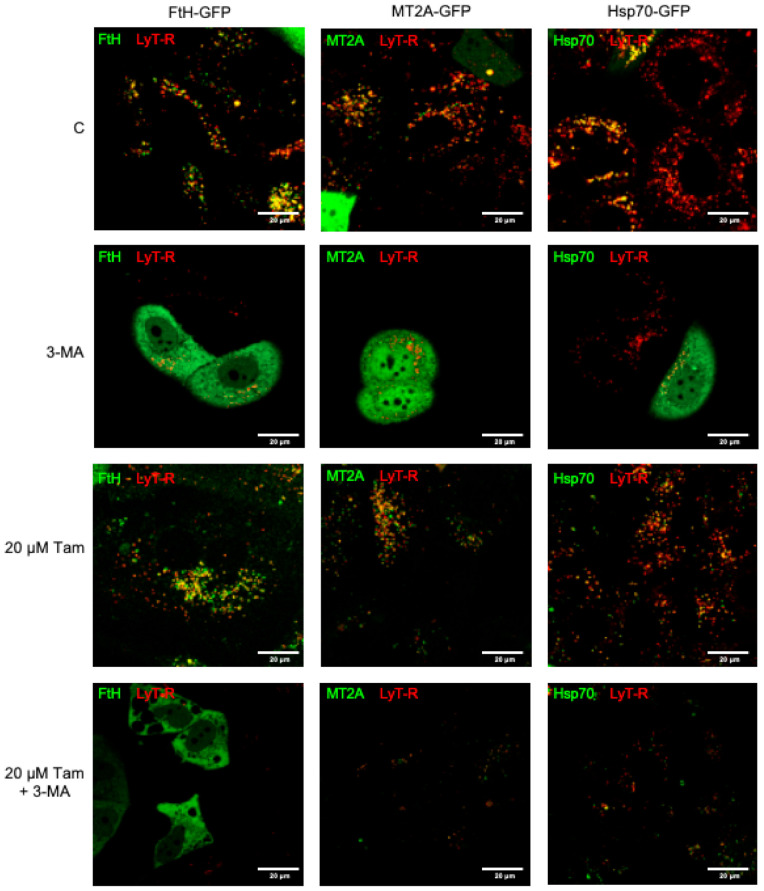
Inhibition of autophagy restores susceptibility to Tam-induced LMP in MCF7 cells overexpressing MT2A, FtH, or Hsp70. Cells transfected as indicated were left untreated (first row) or treated with 5 mM 3-methyladenine (3-MA, 60 min pretreatment before Tam addition; second row), 20 µM Tam for 2 h in the absence or presence of 3-MA (third and fourth row, respectively). After staining with LysoTracker Red as in Figure 2a, cells were imaged by confocal microscopy.

**Figure 8 cancers-13-01252-f008:**
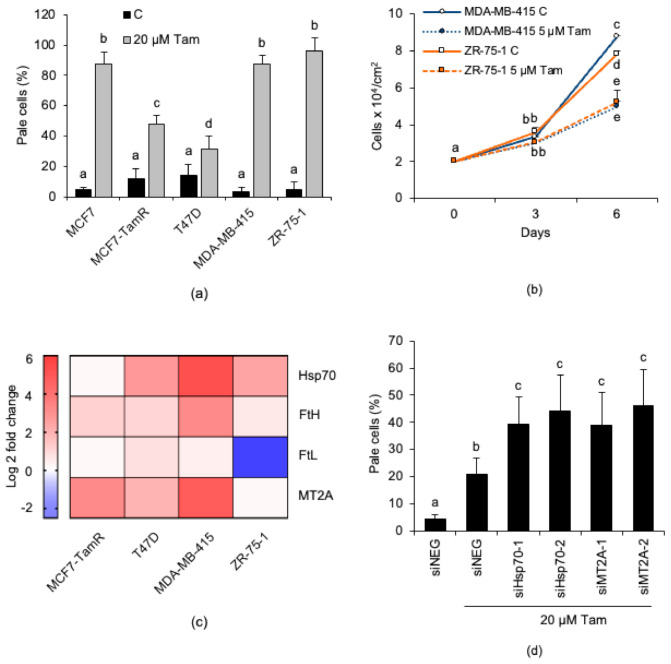
Abundance of iron-binding proteins and capability to activate autophagy concur to shape the susceptibility to Tam-induced LMP of breast cancer cell lines of the luminal A subtype. (**a**) Susceptibility to Tam-induced LMP of breast cancer cells treated with 20 µM Tam for 2 h and subsequently stained with LysoTracker Red for LMP quantification. (**b**) Growth characteristics of MDA-MB-415 and ZR-75-1 cells in the absence or presence of 5 µM Tam. (**c**) Heat map representing the Log2 fold change of the iron-binding protein mRNAs in different breast cancer cell lines; the relative mRNA content was normalized to that of MCF7 cells, which was set to 1; blue or red colors indicate genes that are down- or upregulated, respectively. (**d**) Effect of siRNA-mediated knockdown of Hsp70 and MT2A on susceptibility to Tam-induced LMP in T47D cells. For panels (**a**,**b**,**d**) data and statistical analysis are as in Figure 1.

**Figure 9 cancers-13-01252-f009:**
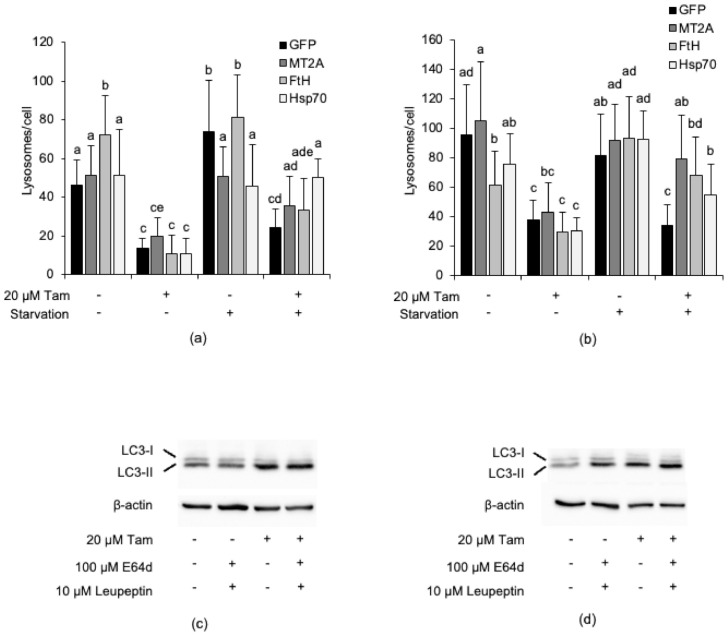
Overexpression of iron-binding proteins and activation of autophagy are both necessary to prevent Tam-induced lysosomal damage in breast cancer cells. (**a**,**b**) Susceptibility of MDA-MB-415 (**a**) and ZR-75-1 (**b**) cells overexpressing GFP, MT2A, FtH, or Hsp70 to LMP induced by treatment with 20 µM Tam for 2 h. (**c**,**d**) Effect of treatment with 20 µM Tam for 2 h on activation of autophagy in MDA-MB-415 (**c**) and ZR-75-1 (**d**) cells transfected as in (**a**,**b**); refer to Appendix A for the original blots. For panels (**a**,**b**), data and statistical analysis are as in Figure 1. Panels (**c**,**d**) are representative of three independent experiments.

**Table 1 cancers-13-01252-t001:** Relative amount of estrogen receptor (ER), Atg protein, and lysosomal protective factor mRNAs in MCF7 and MCF7-TamR cells.

Cell line	ERα	ERβ	Atg5	Atg7	MT2A	FtH	FtL	Hsp70
MCF7	1.0 ± 0.08	1.0 ± 0.15	1.0 ± 0.1	1.0 ± 0.2	1.0 ± 0.4	1.0 ± 0.1	1.0 ± 0.12	1.0 ± 0.05
MCF7-TamR	0.9 ± 0.15	0.9 ± 0.18	0.9 ± 0.4	1.7 ± 0.4	7.6 ± 7.2 *****	2.2 ± 0.5 *****	0.8 ± 0.2	1.1 ± 0.04

Data represent the mean ± SD of three experiments; *: statistical significance (Student’s *t*-test) was set to *p* < 0.05. ERα, ERβ: estrogen receptor α and β; Atg5, Atg7: autophagy-related protein 5 and 7; MT2A: metallothionein 2A; FtH, FtL: ferritin heavy and light chain; Hsp70: heat-shock protein 70.

## Data Availability

Data is contained within the article and Appendix A.

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
