# Peer review of "Autophagy Triggers Tamoxifen Resistance in Human Breast Cancer Cells by Preventing Drug-Induced Lysosomal Damage"

_cancers, 2021, doi:10.3390/cancers13061252_

Round 1
Reviewer 1 Report
The authors have addressed my previous concerns and have improved the overall quality of the study.
Author Response
Response to Reviewer 1.
The authors are sincerely grateful to the Reviewer 1 for the appreciation demonstrated of the revised version of the manuscript.
Reviewer 2 Report
The point 4.2 Viability Assay - the authors should explain better the results obtained.
For all the results that the authors have, the conclusion is too poor. Must be improved.
Author Response
Response to Reviewer 2.
The authors would like to thank the Reviewer 2 for the possibility to better detail part of the results presented in the previous version of the manuscript.
Specific comments
Criticism #1.
The point 4.2 Viability Assay - the authors should explain better the results obtained.
According to the Reviewer request, we have both updated the point ‘4.2 Viability assay’ in Materials and Methods, and also better detailed the presentation of the results achieved using the MTT test to evaluate viability of the cells. The changes to the text have been made using the Word option ‘Track changes’ in the manuscript and are reported below.
Lines 508-514. Materials and Methods, point 4.2 Viability assay
Viability was evaluated with the MTT test. Cells were seeded in 96-well plates at 1.5 x 104 cells/cm2 in 100 µl of growth medium; after treatments, 20 µl of a 5 mg/ml solution of MTT (M2128, 3-(4,5-Dimethyl-2-thiazolyl)-2,5-diphenyl-2H-tetrazolium bromide, Sigma-Aldrich) were added to each well for 2 h at 37 °C under normal growth conditions. The formazan precipitates that form by activity of the mitochondrial dehydrogenases in living cells were dissolved in 100 µl of DMSO by rocking the plates over an orbital shaker for 60 min at room temperature before measurement of the absorbance at 595 nm with the iMark Microplate Reader (Bio-Rad Laboratories).
Lines 169-177. Results: description of Figure 3b.
We next investigated whether the lysosomal effects of Tam were dose-dependent and affected cell viability. We exposed MCF7 cells to increasing concentrations of Tam for 24 h and observed that the number of cells undergoing LMP increases with Tam concentration (Figure 3a). Tam treatment dose-dependently affects viability of MCF7 cells (Figure 3b), but with a kinetic different from LMP development, which always precedes death. Reduction of viability, in fact, is negligible for Tam concentrations up to 15 µM, which, on the other hand, already trigger significant LMP. Cell death becomes highly significant only upon treatment with 20 µM Tam, which reduces the culture viability by about 50%, and can thus be regarded as the LD50 for a 24 h-Tam treatment of MCF7 cells under our experimental conditions.
Lines 200-203. Results: description of Figure 4a.
We found that viability was significantly altered neither at the end of the 2 h of Tam treatment nor at the end of the 24 h recovery in normal growth medium after Tam removal (Figure 4a), and that this did not depend on the absence of LMP (Figure 4b).
Lines 216-219. Results: description of Figure 4g.
As expected, inhibition of autophagy during recovery after Tam treatment significantly reduces cell survival (Figure 4g) and confirms that autophagy plays a prosurvival role in Tam-treated MCF7 cells.
Lines 240-247. Results: description of Figure 5f.
... and that impairing autophagy with a number of validated selective inhibitors restored susceptibility of resistant cells to 5 and 20 µM Tam (Figure 5f). In fact, while did not significantly affect cell viability when used alone, the inhibitors reduced the viability of MCF7-TamR cells in the presence of Tam at both 5 and 20 µM. This finding demonstrates that restraining autophagy restores susceptibility to Tam of otherwise Tam-resistant breast cancer cells. Of interest, the Tam-sensitizing effect of autophagy inhibition, in particular that afforded by CQ and 3MAwas evident already with 5 µM Tam, a concentration to which MCF7-TamR cells are resistant, and, as expected, more pronounced with 20 µM.
Criticism #2.
For all the results that the authors have, the conclusion is too poor. Must be improved.
According to this request, we have expanded the conclusions in order to provide a more comprehensive interpretation of the results presented in the manuscript.
Autophagy plays a relevant role in onset of endocrine resistance of ER+ breast cancer. Although several hypotheses have been proposed, a unifying theory explaining how this occurs at the molecular level is still lacking. Here we show that Tam damages the lysosomal compartment and triggers LMP in all the ER+ breast cancer cell lines tested. Collectively, our data thus demonstrate that LMP may represent an important mechanism by which this anticancer drug exerts its biological action. At the same time, our findings reveal that susceptibility to Tam-induced LMP varies according to both the capability of Tam to activate autophagy in target cells and to the level of cytoplasmic iron-binding proteins. In fact, T47D cells, which overexpress MT2A and Hsp70, are less prone to undergo Tam-induced LMP and cell death compared to MCF7 and the other luminal A cell lines tested. On the other hand, silencing MT2A or Hsp70 in these cells restores LMP, which confirms that these iron-binding proteins act as lysosome-protecting factors that restrain drug-induced lysosomal damage and ensuing cytotoxicity.
However, our findings also reveal that the level of lysosome-protecting factors is not the unique factor that determines the proneness to Tam-induced LMP. MDA-MB-415 and ZR-75-1 cells, which overexpress MT2A or Hsp70 similarly to T47D cells, are susceptible to Tam-induced LMP comparably to MCF7 cells. Analysis of autophagic flux shows that in these cells Tam does not trigger autophagy, which prevents the lysosome-protecting effect brought about by MT2A or Hsp70 overexpression to show up. In keeping with this result, we demonstrate that enforced activation of autophagic flux by starvation restores protection of the lysosomal compartment against Tam-induced LMP. Collectively, our data confirm that the activation of autophagy, which occurs as a consequence of Tam treatment, contributes to Tam resistance of ER+ breast cancer cells by relocating inside the lysosomal lumen the protective factors capable of restraining the drug-induced lysosomal damage.
In conclusion, our results demonstrate that lysosome-protecting factors, such as the iron-binding proteins, by synergizing with activated autophagy might represent an additional risk factor for onset of Tam resistance. According to this view, early identification of breast cancer patients which overexpress such protective factors might help to deploy suitable therapeutic strategies to limit the onset as well as to overcome an already established drug resistance.
